# The limits of action control for deceptive actions in sports: Response inhibition for the basketball pump fake

Carolin Wickemeyer*, Iris Güldenpenning, Matthias Weigelt

Department of Sport & Health, Paderborn University, Paderborn, Germany

* carolin.wickemeyer@uni-paderborn.de

## Abstract

Even NBA players fall for pump fakes in approximately 73% of the time and initiate erroneous blocking movements. To investigate the spatio-temporal and dynamic constraints of inhibition performance in basketball, the basketball-specific anticipation-response inhibition (ARI) task was applied in a quasi-realistic scenario. To this end, a video of a basketball jump shot was presented and participants were instructed to jump up and press a buzzer at the ceiling to "block" the shot (go-trials). In 25% of all trials, a simulation of a pump fake was presented, and participants should withhold their response (stop-trials). To measure response inhibition ability, the point of no return (PNR, signifying a response-inhibition rate of 50%) was calculated. The PNR was located 462 ms before the point of ball release. The response-precision performance improved from the first half of the experiment (Blocks 1–3) to the second half of the experiment (Blocks 4–6), indicating effects of short-term practice. In addition, participants shifted their priority in favor of inhibition after a preceding stop-trial, which is reminiscent of strategic adaptations. Selective biomechanical parameters, measured by a force plate, revealed that response initiation becomes more likely the closer the progression of the execution of the deceptive action moves towards the PNR. Once a response is initiated, it can only be aborted in the early phase of movement execution before the PNR is reached. As a consequence, participants delayed their response in a go-trial, used less force, and slowed down their movements to increase the probability to successfully inhibit the defensive action.

## Introduction

The ability to inhibit a prepared but no longer relevant response is an act of cognitive control and is necessary to adapt quickly to changing circumstances [1]. This is particularly evident in ball games, such as basketball, where the ability to rapidly adjust to changing circumstances is a key element of success. For example, in a one-on-one situation with an attacker and a defender, the defender must anticipate

**Data availability statement:** All relevant data for this study are publicly available from the OSF repository (https://osf.io/hx4nc).

**Funding:** I acknowledge support for the publication cost by the Open Access Publication Fund of Paderborn University with the grant number 20250103 (URL of the funder's website: https://www.ub.uni-paderborn.de/publizieren/open-access/open-access-publikationsfonds).www.ub.uni-paderborn.de/publizieren/open-access/open-access-publikationsfonds).

**Competing interests:** The authors have declared that no competing interests exist.

the attacker's action to successfully defend the attacker. However, if the attacker tries to deceive the defender into a false reaction by providing action-irrelevant (mis-)information, the defender must stop the planned defensive action before its full execution in order to react to the deception in time. One example of such a deception is the pump fake in basketball, which seems to be very effective. A recent study has revealed that approximately 73% of pump fakes executed during National Basketball Association (NBA) games are successful, thereby enhancing offensive scoring [2]. The complete execution of a pump fake has been described as a combination of two actions in a recent taxonomy [3]. The first action is the pump fake as such, where the attacker raises the ball up to a certain point, to mislead the defender into jumping up for a blocking action. This is followed by the attacker stopping the movement and withdrawing the ball. The second action is the actual shot, where the attacker performs the undefended throwing action.

In view of the high probability of successful pump fakes in the NBA [2], it may be that the defensive player recognizes the fake attempt of the attacker during its execution but is unable to stop the prepared but no longer relevant blocking action. As a result, the defender is out of position to block the (real) action, providing the attacker with additional time to perform an unblocked jump shot. How shot deception affects anticipatory behavior in basketball was investigated using video sequences of jump shots and pump fakes, which were occluded at three different points in time: ball at chest, head, and above the head [4]. Results showed that pump fakes became more successful as the throwing sequence reached the end of the movement, for both experienced and novice basketball players. The classification accuracy dropped from 47% at the first to 24% at the third occlusion point. Expert players, who rely on kinematic cues to detect the deception [5], were most susceptible at the final occlusion point, where movement patterns closely resembled a real shot. Furthermore, it was demonstrated that experts mainly focus on the head while defending in a one-on-one situation and only shift their focus to the ball in the throwing phase [6]. Interestingly, this was contrary to another finding [4], which revealed that fixation on the lower body resulted in higher accuracy for distinguishing deceptive from non-deceptive actions.

From these previous studies [2,4,6], which examined pump fake anticipation, the question arises up to which point in time a defensive player can still inhibit the (already prepared) defensive blocking action when confronted with a pump fake by the attacker. In order to investigate this question, a basketball-specific anticipation-response-inhibition task (ARI task) was constructed and validated in two computer-based experiments with a simple finger-lift response in a previous study [7]. In Experiment 2, participants were seated at a desk in front of a computer screen. In 75% of all trials (go-trials), a video of a basketball jump shot was viewed and participants were asked to release the response key to stop the video as precisely as possible at the point where the ball leaves the attackers' fingertips (go-trials). In contrast, a simulated pump fake was presented in 25% of all trials (stop-trials), and participants were asked to inhibit the finger-lift response. To experimentally determine the limits of action control (here, response inhibition), the point of no return (PNR) was calculated, denoting the time point at which the action is irrevocably executed

[8]. The PNR was determined using the 50% probability of successful and unsuccessful response inhibition. In addition, the response-precision performance was measured using the constant error [CE] in the go-trials, which reflects the average directional deviation from the target. Furthermore, response adjustment was evaluated, as indicated by poorer response-precision performance in go-trials following successful and unsuccessful stop-trials. The PNR was 177 ms before the point of ball release, which was consistent with the findings of previous studies using such ARI tasks [9–11]. Further, participants' response-precision performance improved with short-term practice, as participants became better at anticipating the exact point of ball release across the experiment. Furthermore, participants delayed their responses (as signified by a higher CE) in a go-trial following a stop-trial, irrespective of their success in the previous stop-trial. This reflects a strategic adjustment to balance action control for response-precision performance in go-trials and inhibition performance in stop-trials, favoring inhibition over precision.

As real-world defensive actions require the involvement and coordination of multiple degrees of freedom and numerous motor units [12], the findings from the initial study using the basketball-specific ARI task [7] provide only limited evidence concerning the time point up to which a complex action, such as a defender's blocking movement, can be inhibited. This limitation is evident in the majority of research on response inhibition involving only simple button responses [e.g., 11,13–15]. A rare exception that is most relevant for the present study is a previous work in baseball [16] in which the ARI task was used to investigate up to which point in time the batter was able to successfully inhibit a full swing during the flight of the pitch. Specifically, participants were asked to bat against a variety of simulated pitches, some of which would cross the plate inside the strike zone ("strikes") and others outside ("balls"). Each swing was analyzed using motion tracking to determine the response outcome—specifically, whether the batter executed a full swing, completely inhibited the swing, or exhibited a partial or interrupted response. These outcomes were then associated with the timing and nature of the perceptual information available to the batter when attempting to inhibit the action, such as the ball's launch angle immediately after release, early trajectory cues, and late trajectory cues during the ball's flight. The results suggest that batters use different perceptual information about the ongoing throwing action of the pitcher and the ball's trajectory at different times, thereby utilising a stop-signal to inhibit the response action (i.e., batting). The success of this inhibitory action depends critically on the timing of the stop-signal: after a certain "point of no return" in the pitch's trajectory, the batter is no longer able to halt or interrupt the swing, and the swing is inevitably completed.

The present study is conceptually based on previous research [16] and methodologically replicates an earlier investigation [7], focusing on the limits of action control for the basketball pump fake in a quasi-realistic response scenario. To this end, a jump shot was viewed in the go-trials and participants were instructed to jump up and press a buzzer mounted at the ceiling as precise as possible at the exact point in time when the ball leaves the attacker's fingertips. In the stop-trials, a simulated pump fake was viewed, and participants were instructed to inhibit their response. The time at which the pump fake was revealed (i.e., the point at which the video was paused and rewound until the ball was back at hip height) was adjusted based on participants' inhibition performance. During the experiment, participants stood on a force plate, facilitating the examination of different response behaviors (e.g., full response execution, full response inhibition, partial response inhibition).

The study was pre-registered with the following predictions, based on previous studies and theoretical assumptions: First, in Experiment 2 of the initial study using the basketball-specific ARI task, the PNR for a simple motor response (e.g., finger-lift response) was 177 ms [7], which is similar to values reported in other (sport-unspecific) ARI tasks [9–11]. In accordance with the memory-drum theory [17], the simple reaction time (SRT) depends on the complexity of the response and increases with increasing complexity. A more complex response appears to require longer programming and thus, greater storage in the "memory drum". Consequently, it takes a longer time to read from memory before the movement can be initiated. It is therefore hypothesized that if increased response complexity, and thus longer movement-preparation time, affects inhibition performance, the PNR will occur further from the point of ball release for the whole-body response than for the simple finger responses used in the previous study [7] (Hypothesis 1). Second, motor-preparatory activities

(i.e., muscle activations measured by EMG) have been observed in ARI tasks even if participants could successfully inhibit their responses [9,10,18]. These preparatory muscular activations result in a higher proportion of partial and full responses as the stop approaches the target position. Thus, it is hypothesized that if inhibition becomes more difficult the closer the reversal point is to the point of ball release, it will simultaneously result in an increase of erroneous partial and full responses and a decrease of successful inhibitions (Hypothesis 2). Third, as response-precision performance has been demonstrated to be enhanced by short-term practice [7], it is expected that the precision in the second half of the experiment (Blocks 4–6) will be higher (i.e., smaller constant error and more total hits) than in the first half of the experiment (Blocks 1–3) (Hypothesis 3). Fourth, it has been demonstrated that participants strategically adjust their responses based on the previous trial history. This results in a delayed response (i.e., as indicated by a higher CE) in a subsequent go-trial after a previous stop-trial (i.e., go-trial after stop-trial) compared to a previous go-trial (i.e., two consecutive go-trials) [7,19,20]. Most interestingly, in Experiment 2 of the study using the basketball-specific ARI task, this strategic adjustment was found to be independent of whether the previous stop-trial was successfully inhibited or not [7]. If the strategic adaptations observed for simple responses to the basketball pump fake [7] also apply to the whole-body response action in the present experiment, then a higher CE is expected in the subsequent go-trial following a stop-trial than following a go-trial (Hypothesis 4). In addition, it will be examined in an exploratory fashion if the kind of failed response inhibition in a previous stop-trial, as differentiated by an erroneous initial response and an erroneous full response, influences the extent of the CE in the subsequent go-trial.

## Method

The Ethics Committee of the Paderborn University assessed the study as ethically noncritical and granted its approval in written form (42/2023).

### Participants

The recruitment period for the study began on April 4, 2024, and ended on April 22, 2024. All data were saved, analyzed, and published anonymously. The sample size was calculated using MorePower 6.0.4. In the initial study validating the basketball-specific ARI task [7], a main effect for the within-subject factor trial n-1 (successful stop-trial, unsuccessful stop-trial, go-trial) was found for the constant error (i.e., reflecting participants' response-precision performance) in both Experiment 1 ($n_p^2 = .43$) and Experiment 2 ($n_p^2 = .74$). For a conservative sample size estimation, the smallest effect size for the main effect for the within-subjects factor trial n-1 was used, which was $n_p^2 = .43$. Consequently, to achieve a power of .90, a minimum of 18 participants is required for a main effect for the within-subject factor trail n-1 of $f = .869$ and an $\alpha$-value of .05. In total, 21 sport science students (female = 10, male = 11, $M_{age} = 21.43$) participated voluntarily for course credit but no financial reward. Sixteen participants declared themself as being right-handed, the other 5 participants as being left-handed. Prior the experiment, participants signed an informed consent form. Normal or corrected visual acuity, no physical impairment, or psychopathological and neurological disorders nor basketball experience (on a club level) were mandatory to participate. However, some participants had gained practical experience at school or during a basketball course within the sport science study program.

### Apparatus and stimuli

The stimulus material used in the present study was a video of a basketball jump shot (see Fig 1) viewed from a front-view perspective. At the onset of the video, the basketball player faces the camera with the ball centered in front of his body (Fig 1, first image on the far left). Thereafter, the ball was bounced three times with the right hand (the bouncing was completed 1850 ms after video onset). Subsequently, the basketball player bends his knees and hips, positions his throwing hand behind the ball, and raises his arm until the back of his right hand is above his forehead (with the ball positioned above his forehead at 2400 ms after video onset). This is followed by the execution of

 

**a**

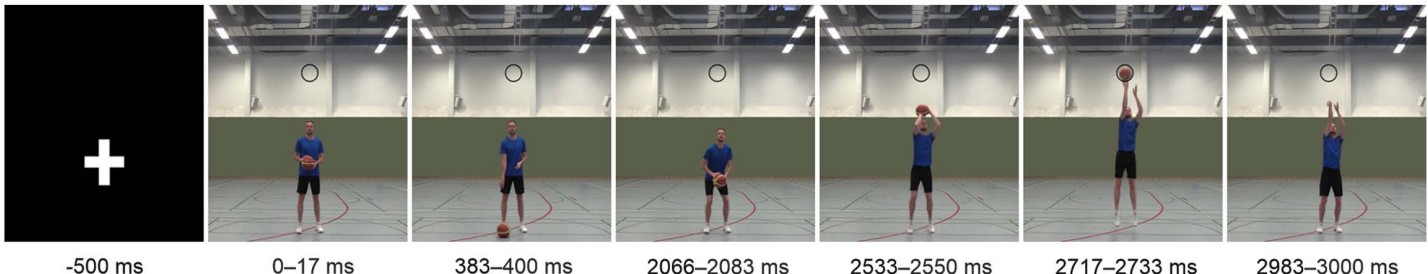

**b**

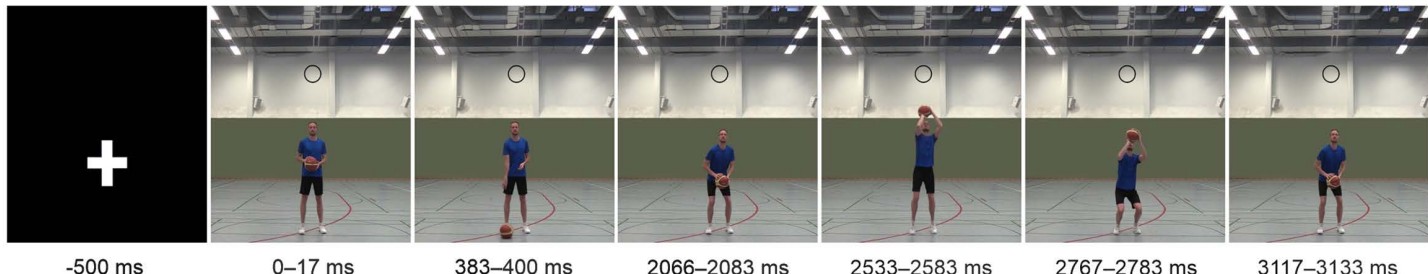

**Fig 1. Still frames taken as examples from the video stimulus.** a) Basketball jump shot used in the go-trials. b) Basketball pump fake used in the stop-trials.

a vertical jump, with the shoulder of the throwing hand being lifted. The ball left the fingertips of his right hand at the highest point of the jump (2717 ms after video onset), with his throwing arm directed towards the basket (so-called follow-through). This specific point in time served as the target for the present experiment, which participants were required to anticipate for their response. For a more detailed description of the stimulus material, please refer to the previous paper [7]. The video stimulus was displayed at the opposite side of the room on a wall with a beamer (Optoma CinemaX D2 HDMI RJ-45) with a 4K UHD resolution, and a refresh rate of 60 Hz. The size of the presented video was 150 x 275 cm. The presentation of the image sequence was controlled by ©Presentation 24.0. The single video images were updated every refresh rate (60 Hz = 16.7 ms). Two buzzers were attached to the ceiling above the participants, and one buzzer was placed on the table in front (see Fig 2). In order to standardize the jump height, participants were asked to stand up straight and extend their dominant hand towards the corresponding left or right ceiling buzzer before the experiment. The height of the ceiling buzzer was adjusted, so that the participant's finger-tips lightly touched the buzzer from below. The measurement of ground reaction forces was facilitated by employing an AMTI force plate (model: BP600900-1K-CTT). Force plate signals were processed using a six channel AMTI GEN 5 Amplifier.

## Procedure

Prior to the experiment, a brief questionnaire was completed by the participants to collect demographic data and sport-specific information. The questionnaire inquired about age, sex, corrected visual acuity, psychopathological and neurological disorders [21], basketball expertise, the sport being practiced, and the training frequency in hours per week. The instructions for the experimental task were provided in both written and verbal form. To this end, participants were

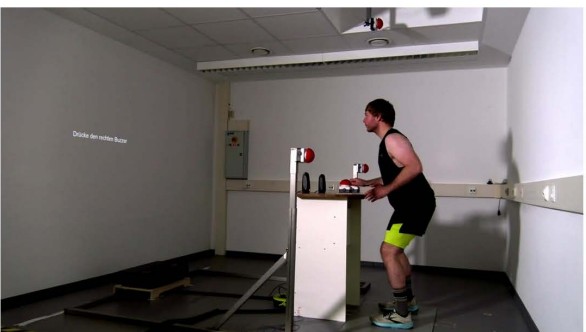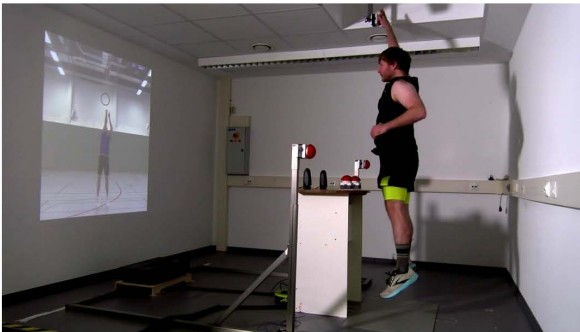

**Fig 2. The experimental setup.** The starting position (left picture) of the participants at the beginning of each trial and the final position (right picture) of the go-trial is shown.

asked to stand on the force plate in a defensive position, with their feet placed parallel and shoulder-width apart, their knees slightly bent, and with an upright torso. Their arms should be raised at the sides with the elbows slightly bent. The participants were asked to return to this defensive position after each trial (see Fig 2). To start a trial, they were instructed to press the buzzer on the table in front of them with their dominant hand. After pressing (and holding down) the buzzer for 500 ms, a fixation cross was displayed for another 500 ms, before the video started. Throughout the experiment, two different trials were randomly presented. In 75% of all trials, a basketball jump shot was presented and participants were asked to "block" the jump shot by releasing the table buzzer, jumping up, and pressing the buzzer on the ceiling with their dominant hand at the time point when the ball left the attacker's fingertips (go-trials). Subsequent to the pressing of the ceiling buzzer, the video stopped immediately. In 25% of all trials, a basketball pump fake was simulated by playing the video up to a certain point in time before the ball left the fingertips, then pausing the video for 50 ms, before playing it back (rewinding) until the ball was at hip height again (stop-trials). In these stop-trials, participants were instructed to withhold their response, while keeping the table buzzer pressed down.

The point in time, at which the video was stopped, was adjusted by a staircase-tracking algorithm [22] based on participants' performance in the previous stop-trial, with a fixed step size of 16.7 ms, which corresponded to one video frame. If a participant successfully inhibited the buzzer lift-response in the previous stop-trial, the stop-signal interval between the point of ball release and the video stop was decreased for the next stop-trial by one video frame (i.e., the stop-signal interval became smaller and thus, the video stop-signal moved closer to the point of ball release). If the participant was not able to successfully inhibit the buzzer lift response, the stop-signal interval was increased for the next stop-trial (i.e., the stop-signal interval became larger and thus, the video stop-signal moved away from the point of ball release). The initial stop-signal was set at 516 ms before the point of ball release based on previous pilot testing.

The whole experiment consisted of 240 trials, equally divided into six blocks. The trials varied with regard to the type of trial (go-trial vs. stop-trial) and were presented in a randomized order, generated by the ©Presentation software. To avoid participants' strategy of waiting for stop-trials and to increase readiness to respond to go-trials for increased response-precision performance, a high percentage of go-trials (75%) was chosen [23]. Thus, each test block consisted of 30 go-trials and 10 stop-trials. At the end of each block, participants received feedback about their overall performance [23], including (1) the mean anticipation-response accuracy, provided as constant error, reflecting the average directional error relative to the time point of ball release, (2) the mean probability of responding in a stop-trial, and (3) the total number of missed go-trials (go-omissions). After each block, participants were allowed to take a break to maintain concentration and minimize fatigue (due to the jump and reach movement).

 

To become familiar with the task, two practice blocks of 40 trials each were completed before the experiment started. Practice Block 1 consisted of go-trials only. This allowed participants to become familiar with the task and ensured that they responded precisely by knowing when and how to jump off to press the buzzer at the ceiling. In Practice Block 2, stop-trials were added. In both practice blocks, participants received additional feedback about their response-precision performance (i.e., constant error). In Practice Block 2, they also received feedback about their correctness of their response (correct inhibition/ incorrect inhibition) 1000 ms after every trial for 2000 ms. After the practice blocks, participants only received feedback after a trial if they had responded before the video began or if they did not respond in the go-trials, to minimize missed trials. Further details regarding the feedback can be found in the descriptions of the previous study [7]. The entire experiment lasted about 70–90 minutes.

## Statistical analysis

To determine the inhibition performance, the PNR was calculated by averaging the peaks and valleys of all runs for each participant over all six testing blocks [24]. A run was defined as a series of increasing or decreasing stop-signal intervals in one direction. Thus, a peak is defined as the last stop-signal value in an ascending series of stop-signal intervals before the participant can suppress the response again and the stop-signal interval decreases again. A valley is the last stop-signal value in a descending series of stop-signal intervals before the participant can no longer suppress the response, and the stop-signal interval increases (again). For a failed inhibition in a stop-trial, a distinction is made between full responses and partial responses. For each stop-signal interval, the frequency of lifting the hand off the buzzer (erroneous initial response), lifting the hand off the buzzer and jumping off the ground (erroneous partial responses), and jumping off the ground and pressing the buzzer at the ceiling (erroneous full response) were determined. Trials in which participants lifted their hand off the table buzzer before the video started or did not respond were excluded (3.6% of total trials).

For response-precision performance, the constant error (CE) was calculated by subtracting the target (i.e., the time point when the ball left the attacker's fingertips) from the time point when the participant pressed the ceiling buzzer [CE = (X – 2717 ms) ms]. The CE was recorded as positive if the participants pressed the ceiling buzzer after the ball had left the attacker's fingertips and thus, responded too late. Conversely, the CE was recorded as negative if the participants pressed the ceiling buzzer before the ball left the attacker's fingertips and thus, responded too early. Total hits (TH) were defined as the number of correct responses (i.e., pressing the ceiling buzzer at the exact time point when the ball leaves the attacker's hand). Furthermore, the buzzer-release time (BRT) was calculated by taking the mean time from lifting the hand off the table buzzer relative to the target. In addition, the time when participants left the ground in relation to the target was calculated to determine the take-off time. The take-off time (TOT) was defined as the point in time where the force curve during take-off becomes lower than the participant's body weight.

For the PNR, CE, TH, BRT, and TOT, mean values were calculated for the first half (Blocks 1–3) and second half (Blocks 4–6) of the experiment, as well as for the whole experiment, respectively. Possible changes in behavior between the two halves of the experiment (indicating effects of short-term practice) were evaluated using standard t-tests. To identify possible strategic adjustments after a stop-trial, a repeated measures ANOVA was conducted, and the Greenhouse-Geisser adjustment was used for correction if any violations of sphericity occurred. To this end, the dependent variables constant error (CE), buzzer-release time (BRT), and take-off time (TOT) were analyzed regarding the within-subject factor trial n-1 (go-trial, successful inhibition in a stop-trial, unsuccessful inhibition in a stop-trial). To investigate differences in force-time profiles, repeated measures ANOVAs were conducted. The analyses focused on key movement time points (t0–t3) and the timing and magnitude of peak vertical force. Separate ANOVAs were run for each dependent variable, with either trial type (go-trial, successful stop-trial, unsuccessful stop-trial) or response-precision performance (hit, too late, too early) as the within-subject factor. Additionally, force plate data was used to analyse the unsuccessful inhibitions in more detail, and the dependent variables CE, BRT, and TOT were analyzed regarding the within-subject factor trial n-1 (go-trial, successful inhibition, erroneous initial response, and erroneous full

response stop-trial) (see S1 Appendix). All statistical analyses were performed using ©IBM SPSS Statistics 29. For all analyses, the *a*-value was set to .05, and partial eta-square was used to calculate the effect size. Post-hoc t-tests were conducted in case of significant main effects.

## Results

### Inhibition performance

On average, participants were able to inhibit their response on average in 50.7% of all stop-trials, indicating that the staircase-tracking algorithm was applied successfully. Throughout the experiment, the video was reversed between 317 and 667 ms before the ball was released. Since the stop-signal interval between the premature stop and the reverse of the video and the point of ball release was adjusted based on participants' individual performance, most of the stop-signals were around the mean PNR (see Fig 3). The mean PNR was located 461.7 ms (SEM = 11.07 ms; Table 1 and Fig 4) before the target and remained consistent throughout the experiment. The PNR did not improve from the first half of the experiment (Blocks 1–3) to the second half of the experiment (Blocks 4–6) (*p* = .524).

### Response-precision performance

Participants were able to anticipate the exact point where the ball left the attacker's fingertips in 6.4 (SEM = 1.5, Table 1) go-trials out of 180 total go-trials (total hits [TH]; i.e., CE = 0). The number of hits was lower in the first half of the experiment (Blocks 1–3: 2.5 TH [SEM = .65]) than in the second half of the experiment (Blocks 4–6: 3.9 TH [SEM = .93]) [*t*(20) = −2.430, *p* = .025, *d* = −.530].

On average, participants pressed the ceiling buzzer about 89.8 ms (SEM = 6.93 ms) after the ball left the attacker's fingertips (Blocks 1–6; see Fig 5). The mean CE was higher in the first half of the experiment (Blocks 1–3: 94.4 ms, SEM = 6.84 ms) compared to the second half of the experiment (Blocks 4–6: 85.4 ms, SEM = 7.62 ms) [*t*(20) = 2.192, *p* = .04; *d* = .478].

The mean BRT was 219.7 ms (SEM = 9.67, Table 1) before the target and was significantly closer to the target in the first half (214.1 ms) than in the second half (225.2 ms) of the experiment [*t*(20) = −3.144, *p* = .005, *d* = −.686]. The mean

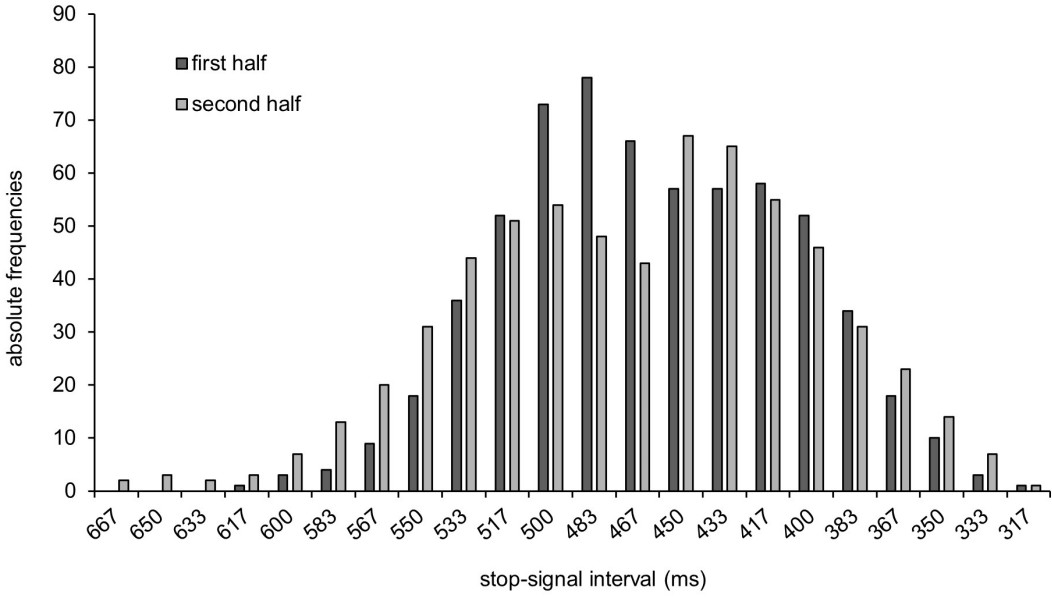

**Fig 3. The (absolute) frequency distributions of stop-signal intervals.**

**Table 1. Means and standard errors (in brackets) for response-precision-performance and inhibition performance.**

| inhibition performance | | | | |
|---|---|---|---|---|
| | PNR (ms) | p(respond/signal) (%) | | |
| first half | 464.7 (12.87) | 47.9 (0.01) | | |
| second half | 460.2 (10.57) | 50.8 (0.11) | | |
| overall | 461.7 (11.07) | 49.3 (.005) | | |
| **response-precision performance** | | | | |
| | CE (ms) | hits (total) | movement time (ms) | take-off time (ms) |
| first half | 94.4 (6.84) | 2.5 (.65) | 214.1 (9.45) | 50.9 (9.75) |
| second half | 85.4 (7.62) | 3.9 (.93) | 225.2 (10.19) | 59.9 (9.42) |
| overall | 89.8 (6.93) | 6.4 (1.5) | 219.7 (9.67) | 55.4 (9.32) |
| **post-stop-trial adjustments (ms)** | | | | |
| | go-trial/ go-trial | successful stop-trial/ go-trial | unsuccessful stop-trial/ go-trial | |
| **CE** | | | | |
| first half | 88.9 (7.01) | 107.5 (6.79) | 116.9 (8.19) | |
| second half | 80.2 (7.81) | 95.8 (8.60) | 105.4 (8.65) | |
| overall | 84.5 (7.06) | 101.8 (7.23) | 109.9 (7.33) | |
| **take-off time** | | | | |
| first half | 56.8 (10.41) | 35.1 (8.25) | 28.2 (10.27) | |
| second half | 65.0 (9.77) | 47.7 (9.49) | 39.5 (9.6) | |
| overall | 61.0 (9.79) | 41.0 (8.32) | 34.9 (8.96) | |
| **movement time** | | | | |
| first half | 217.3 (9.91) | 229.3 (10.57) | 199.1 (10.72) | |
| second half | 229.3 (10.57) | 204.3 (8.39) | 207.2 (11.28) | |
| overall | 223.3 (10.04) | 210.7 (8.62) | 204.1 (9.99) | |

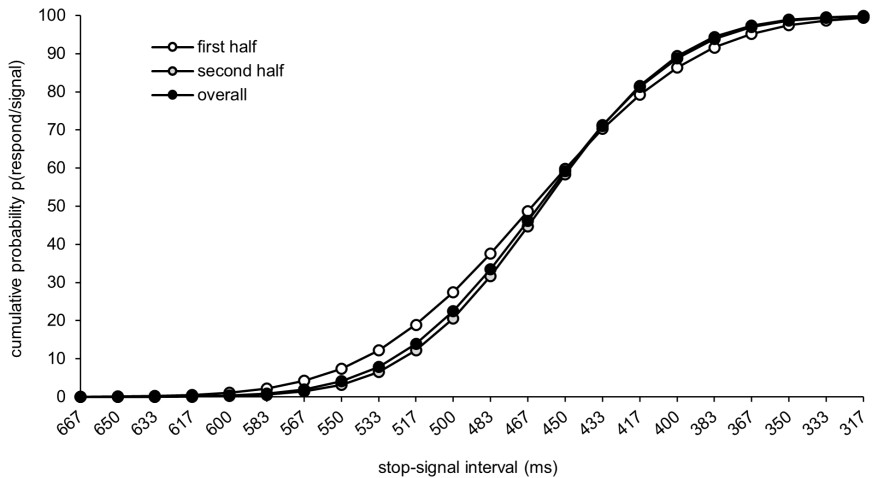

**Fig 4. The cumulative functions of the probability to respond in a stop-trial.**

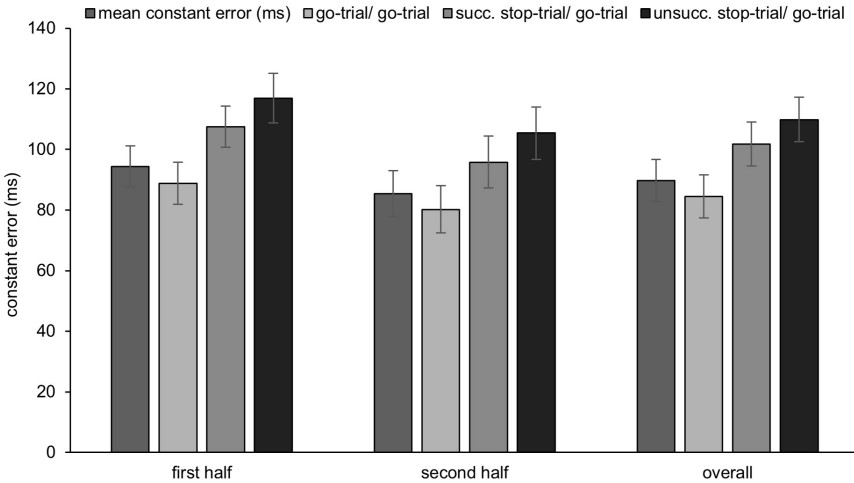

**Fig 5. The mean constant errors (CE) and post-stop-trial adjustments.**

TOT was 55.4 ms (SEM = 9.32) before the target and participants jumped at the same time in the first (50.9 ms) compared to the second (59.9 ms) half of the experiment ($p = .054$).

## Post-stop-trial adjustments

The mean CE of the present go-trial after a previous go-trial was 84.5 ms (SEM = 7.06), was 101.8 ms (SEM = 7.23) after a successful stop-trial, and was 109.9 ms (SEM = 7.33) after an unsuccessful stop-trial. For the CE, the ANOVA showed a main effect for the factor trial n-1 [$F_{(2, 40)} = 22.245$, $p < .001$, $n_p^2 = .527$]. The precision in two consecutive go-trials was higher than after a successful stop-trial [$t_{(20)} = −3.684$, $p = .001$, $d = −.804$] and an unsuccessful stop-trial [$t_{(20)} = −8.425$, $p < .001$, $d = −1.838$]. Moreover, the precision after a successful stop-trial was higher than after an unsuccessful stop-trial [$t_{(20)} = −2.135$, $p = .045$, $d = −.466$].

Participants released the table buzzer (BRT) in the present go-trial 223.3 ms (SEM = 10.04) before the target after a previous go-trial, 210.7 ms (SEM = 8.62) after a successful stop-trial, and 204.1 ms (SEM = 9.99) after an unsuccessful stop-trial. The ANOVA on the BRT revealed a main effect of the factor trial n-1 [$F_{(2, 40)} = 9.658$, $p < .001$, $n_p^2 = .326$]. The BRT in two consecutive go-trials was earlier than after a successful stop-trial [$t_{(20)} = 2.328$, $p = .031$, $d = .508$] and after an unsuccessful stop-trial [$t_{(20)} = 4.850$, $p < .001$, $d = 1.058$]. Whether the participants could successfully inhibit their response in a previous stop-trial or not, did not affect the BRT ($p = .094$).

Participants left the ground (TOT) in the present go-trial 61.0 ms (SEM = 9.79) before the target after a previous go-trial, 41.0 ms (SEM = 8.32) after a successful stop-trial, and 34.9 ms (SEM = 8.96) after an unsuccessful stop-trial. The ANOVA on the TOT identified a main effect of the factor trial n-1 [$F_{(2, 40)} = 16.489$, $p < .001$, $n_p^2 = .452$]. The TOT in two consecutive go-trials was earlier (with regard to the target) than after a successful stop-trial [$t_{(20)} = 3.430$, $p = .003$, $d = .748$] and after an unsuccessful stop-trial [$t_{(20)} = 6.480$, $p < .001$, $d = 1.414$]. Whether the participants could successfully inhibit their response in a previous stop-trial or not, did not affect the TOT ($p = .160$).

## Ground reaction forces

To obtain a more in-depth understanding of the inhibition of complex movements, the vertical ground reaction force (Fz) was measured by implementing a force plate on which the participants stood during the entire experiment. The vertical force-time profile of a response was similar to a countermovement jump (CMJ) (Fig 6) [25]. But in contrast to a CMJ, the response

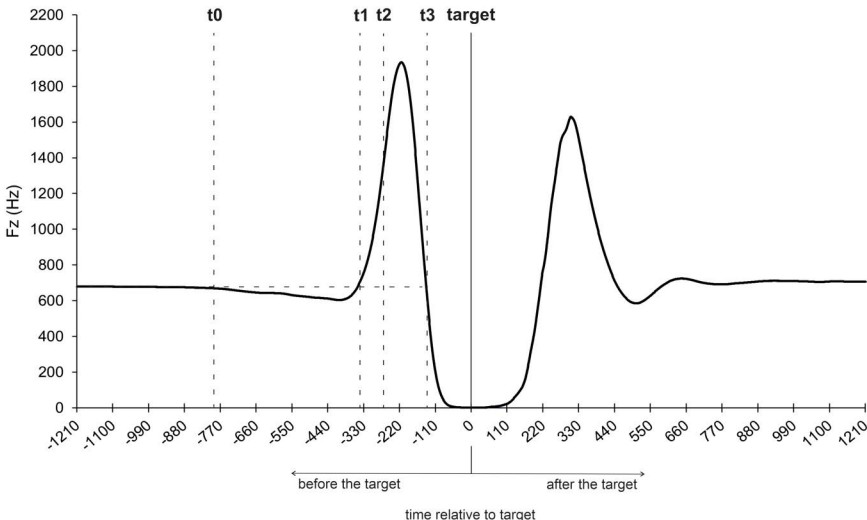

**Fig 6. Vertical force-time profile of go-trials in which participants hit the target.**

movement in the experiment was not initiated from a completely upright position, but from a static defensive position in which the knees and hips were slightly bent (see Fig 2). To estimate the characteristic phases of the CMJ, the mean body weight (BW) was calculated by averaging the force of the static defense position (weighing phase) [26]. The response movement was initiated by a downward movement with a descending acceleration of the center of mass (COM) (t0–t1), whereby the ground reaction force decreases compared to a threshold value. The threshold value corresponds to five times the standard deviation of the BW [26]. At time point t1, the highest descending velocity of the COM is reached, and the downward movement must be decelerated by eccentric work of the leg muscles (t1–t2). At time point t2, the COM is at its lowest point. This is followed by the concentric extension of the ankle, knee, and hip joints, which begins with a force greater than the weight force, the so-called initial force (at t2), as a result of the deceleration movement. At time point t3, the vertical force drops below five times standard deviation of the BW, and the participant leaves the floor. In Fig 6, the average movement response and the relative time points (t0–t3) are visualized by the force-time-profile of the responses for those go-trial, in which participants responded exactly on target, representing the ideal force-time-profile of a CMJ.

### Trial type

To statistically examine differences in the force-time profile with regard to the trial type (see Fig 7), several repeated measures ANOVAs were conducted. Analyses focused on the characteristic points of the movement (t0–t3) as well as on the timing and magnitude of the maximum vertical force (i.e., the highest point of the curve between t2 and t3). These variables were examined with respect to the within-subject factor trial type (go-trial, successful inhibition, unsuccessful inhibition).

Responses were initiated (t0) 792 ms before the target in a go-trial, 648 ms for successful inhibitions in a stop-trial, and 719 ms for unsuccessful inhibitions in a stop-trial. A repeated measures ANOVA with the within-subject factor trial type (go-trial, successful stop-trial, unsuccessful stop-trial) was conducted and a main effect was found [$F(1.446, 27.926)$ = 81.013, $p < .001$, $n_p^2 = .802$]. The responses were initiated earlier in the go-trial compared to the successful stop-trials [$t(20)$ = 3.895, $p < .001$, $d = .850$], but at a similar time compared to the unsuccessful stop-trials ($p = .244$). In the successful stop-trials, the responses were initiated at a similar time compared to the unsuccessful stop-trials ($p = .167$).

The highest descending velocity (t1) was reached 297 ms before the target in the go-trials, 246 ms in the successful stop-trials, and 317 ms in the unsuccessful stop-trials. There was a main effect for the factor trial type [$F(1.371, 27.427)$ =

98.370, $p < .001$, $n_p^2 = .831$]. T1 was earlier in the go-trials compared to the successful stop-trials [$t(20) = 8.148$, $p < .001$, $d = 1.778$], but later compared to unsuccessful stop-trials [$t(20) = -6.759$, $p < .001$, $d = -1.475$]. In the unsuccessful stop-trials, t1 was reached earlier compared to the successful stop-trials [$t(20) = -12.208$, $p < .001$, $d = -2.664$].

The COM is at its lowest point (t2) 214 ms before target in the go-trials, 92 ms in the successful stop-trials, and 232 ms in the unsuccessful stop-trials. Four participants were excluded from further analysis as the data was not sufficient to calculate t2 for a successful stop-trial. There was a main effect for the factor trial type [$F(1.051, 16.821) = 37.677$, $p < .001$, $n_p^2 = .702$]. T2 was earlier in the go-trials, compared to the successful stop-trials [$t(16) = 5.622$, $p < .001$, $d = 1.364$], but later compared to the unsuccessful stop-trials [$t(16) = -4.623$, $p < .001$, $d = -1.121$]. T2 was reached earlier in the unsuccessful stop-trials, compared to the successful stop-trials [$t(16) = -6.690$, $p < .001$, $d = -1.622$] (Fig 7).

The maximum vertical force was reached 148 ms before the target in the go-trials, 145 ms in the successful stop-trials, and 183 ms in the unsuccessful stop-trials. The repeated measures ANOVA showed a main effect for the factor trial type [$F(1.142, 22.860) = 23.629$, $p < .001$, $n_p^2 = .542$]. The maximum vertical force in the go-trial was reached at a similar time compared to the successful stop-trials ($p = .679$), but significantly later compared to the unsuccessful stop-trials [$t(20) = -14.288$, $p < .001$, $d = -3.118$]. The maximum vertical force was reached earlier in the unsuccessful stop-trials compared to the successful stop-trials as well [$t(20) = -5.550$, $p < .001$, $d = -1.211$].

The maximum vertical force was 1530 N in the go-trials, 866 N in the successful stop-trials, and 1424 N in the unsuccessful stop-trials. There was also a main effect for the factor trial type [$F(2, 40) = 122.313$, $p < .001$, $n_p^2 = .859$]. The maximum vertical force in the go-trials was higher compared to the successful stop-trials [$t(20) = 15.284$, $p < .001$, $d = 3.335$], and higher compared to the unsuccessful stop-trials [$t(20) = 2.193$, $p = .040$, $d = .479$]. The maximum vertical force in the successful stop-trials was smaller compared to the unsuccessful stop-trials [$t(20) = -12.399$, $p < .001$, $d = -2.706$] (Fig 7). The last important time point t3, which represents the take-off time, was not analyzed since participants did not jump off the ground, when they were able to inhibit their response in a stop-trial.

**Response-precision performance**

Differences in the force-time profile according to response-precision performance (hit, too late, too early) (Fig 8) were analyzed using repeated measures ANOVAs. The analyses considered the movement's characteristic points

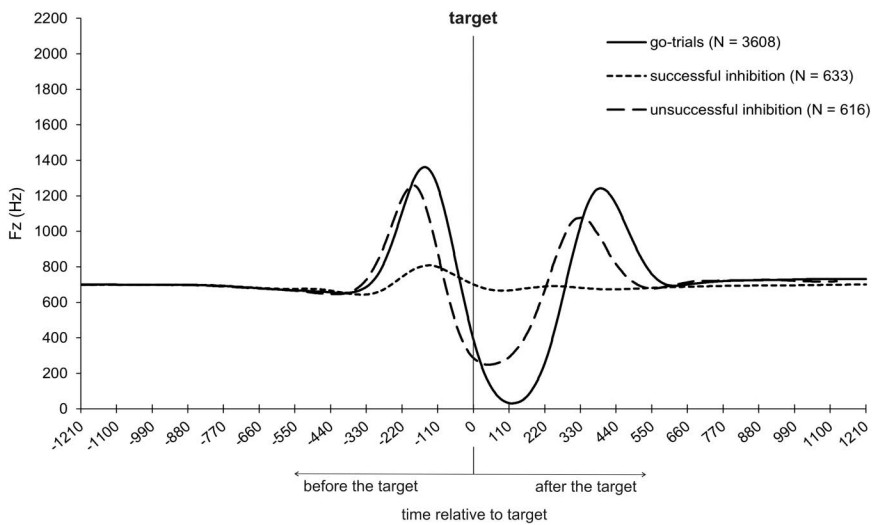

**Fig 7. Vertical force-time profiles of go-trials, successful stop-trials, and unsuccessful stop-trials.**

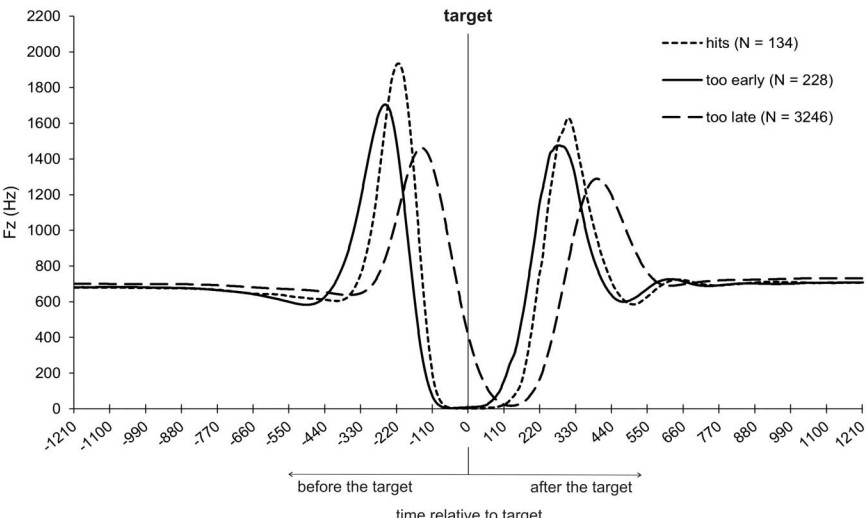

**Fig 8. Vertical force-time profiles of hits, too early responses, and too late responses.**

(t0–t3) and both the timing and magnitude of the maximum vertical force (i.e., the peak between t2 and t3). Four participants were excluded from these analyses because they did not produce responses on target and/or too early responses.

When participants responded exactly on target, they initiated the response (t0) on average 666 ms before the point of ball release. If they responded too early, they initiate their response on average 757 ms and if they responded too late 722 ms before the point of ball release. A repeated measures ANOVA for the factor response-precision performance showed that the initiation of the responses regardless of precision were similar ($p = .355$).

The highest descending velocity (t1) was reached 343 ms before the target, when participants responded on target, 395 ms before the target, when participants responded too early, and 285 ms before the target, when participants responded too late. There was a main effect for the factor response-precision performance [$F(1.329, 21.264) = 84.663$, $p < .001$, $n_p^2 = .841$]. T1 was significantly later when participants responded on target, compared to too early responses [$t(16) = -5.620$, $p < .001$, $d = -1.363$], and was significantly earlier, compared to too late responses [$t(16) = 12.316$, $p < .001$, $d = 2.987$]. When participants responded too early, t1 was significantly earlier compared to too late responses [$t(16) = 10.726$, $p < .001$, $d = 2.601$].

The COM was at its lowest point (t2) 276 ms before the target, when participants responded on target, 324 ms when they responded too early, and 207 ms when they responded too late. There was a main effect for the factor response-precision performance [$F(1.291, 20.661) = 88.699$, $p < .001$, $n_p^2 = .847$]. When participants responded on target, t2 was significantly later than for too early responses [$t(16) = -4.786$, $p < .001$, $d = -1.161$] and was significantly earlier than for too late responses [$t(16) = 15.164$, $p < .001$, $d = 3.678$]. If participants responded too early, t2 was significantly earlier than for too late responses [$t(16) = 11.148$, $p < .001$, $d = 2.704$].

The maximum vertical force was reached 224 ms before the target, when participants responded on target, 261 ms before the target, when they responded too early, and 148 ms before the target, when they responded too late. There was a main effect for the factor response-precision performance [$F(2, 32) = 219.735$, $p < .001$, $n_p^2 = .932$]. When participants responded on target, the maximum vertical force was reached later compared to too early responses [$t(16) = -7.114$, $p < .001$, $d = -1.725$] and was reached earlier compared to too late responses [$t(16) = 17.283$, $p < .001$, $d = 4.192$]. When

participants responded too early, the maximum force was reached earlier than for too late responses [$t(16)$ = 20.525, $p < .001$, $d = 4.978$].

The maximum force was 2063 N, when participants responded on target, 1946 N, when they responded too early, and 1574 N, when they responded too late. There was a main effect for the factor response-precision performance [$F(1.500, 24.000)$ = 40.932, $p < .001$, $n_p^2 = .719$]. The maximum force was higher for responses on target compared to too early responses [$t(16)$ = 2.389, $p < .001$, $d = .580$], but it was higher for responses on target compared to too late responses [$t(16)$ = 6.901, $p < .001$, $d = 1.674$]. The maximum force was also higher when participants responded too early than when they responded too late [$t(16)$ = 8.032, $p < .001$, $d = 1.948$] (Fig 8).

The TOT (t3) was 142 ms before the target, when participants responded on target, 175 ms before the target, when they responded too early, and 43 ms before the target, when they responded too late. There was a main effect for the factor response-precision performance [$F(2, 32)$ = 245.345, $p < .001$, $n_p^2 = .939$]. The TOT was significantly later when participants responded on target, as compared to too early responses [$t(16)$ = −7.093, $p < .001$, $d = −1.720$], and significantly earlier, compared to too late responses [$t(16)$ = 14.142, $p < .001$, $d = 3.430$]. The TOT was significantly earlier for too early responses compared to too late responses [$t(16)$ = 19.863, $p < .001$, $d = 4.588$] (Fig 8).

## Discussion

### Inhibition performance

The results for inhibition performance show that the PNR for the whole-body defensive blocking action was 462 ms before the point of ball release. This is of theoretical relevance, as the PNR for this more complex response occurred considerably earlier than for the less complex finger-lift response in the ARI task, which was 177 ms in Experiment 2 of the previous study [7]. However, in order to compare the inhibition ability for these two response actions of different complexity, two aspects obscure the results for the PNR observed in the present experiment. The first aspect is less obvious and relates to the response-precision performance, as indicated by the CE. In both studies, participants responded too late on average, but to a different extent. While the CE was 16 ms in the ARI task in Experiment 2 in the initial study [7], it was 90 ms in the present study. These values must be added to the observed PNR, resulting in 193 ms and in 552 ms for the PNR proper, respectively. The second aspect is more obvious and relates to the movement time following response initiation, which needs to be subtracted from the PNR proper. While the movement time for the finger-lift response in the initial study [7] is negligible, it must be corrected for in the present experiment. The response movement began when participants lifted their hands off the table buzzer. Therefore, the mean buzzer-release time (BRT) of 220 ms was subtracted from the PNR proper, resulting in a "corrected" PNR proper of 332 ms before movement initiation. Thus, the "corrected" PNR proper for the complex whole-body response action in the present study was earlier than the PNR proper for the simple finger-lift response (193 ms) in the initial study [7]. Therefore, Hypothesis 1 can be confirmed. The "memory-drum theory" [17] provides an explanation for the differences in the PNR between the two response actions of different complexity.

According to this theory [17], simple reaction time depends on the complexity of the response. More complex response actions require larger motor programs and, consequently, greater storage in the "memory drum". For the present study, this means that motor planning processes for the complex whole-body response action required a longer time to read from memory before movement initiation. Thereby, the motor planning processes not only seem to be concerned with the initiation of the response action, but also with the time it takes to execute the movement. This is supported by the fact, that the PNR proper of 552 ms was greater than the "corrected" PNR proper of 332 ms. It suggests that participants anticipated the time it would take them to lift their hand off the table buzzer, jump off the ground, and hit the ceiling buzzer when preparing for the whole-body response action. This also explains why the PNR proper moved even farther away from the anticipated target time in the present experiment. If participants had not accounted for movement time, they would have been always too late without any chance to "block" the jump shot of the attacker.

This assumption is supported by previous results from a baseball-batting task, which also required a whole-body response action and reported a mean duration of 212 ms between the onset of the swinging and the onset of the stopping phase [16]. This temporal span is remarkably similar to the results of the present study (i.e., a BRT of 220 ms). It suggests that inhibition performance may be similar across different sports relying on anticipation skills, even if these support such different tasks as batting a ball in baseball and blocking a ball in basketball.

Besides identifying the PNR for the present ARI task, it was of interest to examine participants' inhibition performance when the stop-signal moves closer to the target. Here, it was assumed that response inhibition becomes more difficult the further the pump-fake movement displayed on the video progresses towards the point of ball release. As a result, it was expected that the number of erroneous partial responses and erroneous full responses would increase (Hypothesis 2).

The results showed that as the reversal point of the video approached the point of ball release, the successful inhibition of the defensive response action during stop-trials decreased. While this initially (almost) only triggered erroneous initial responses (i.e., participants only lifted their hand off the table buzzer but did not jump off the ground). The number of erroneous partial responses (i.e., participants jumped off the ground without hitting the ceiling buzzer) and erroneous full responses (i.e., participants jumped off the ground and hit the ceiling buzzer) started to increase around the PNR proper. Conversely, the number of erroneous initial responses decreased during this time (see Fig 5). In other words, the number of inappropriate defensive response actions increased as the stop-signal interval approached the point of ball release. This may be explained with motor-preparation activities, which entail the activation of muscle groups involved in movement execution. As demonstrated in ARI tasks with simple finger-lift responses, muscle activity can be observed even when participants could successfully inhibit their response [9,10]. For the present task, it can be concluded that once the amount of muscle activity during motor preparation surpasses a certain threshold, the movement response can no longer be suppressed. At this point, the participant will inevitably initiate the inappropriate defensive blocking action.

### Response-precision performance

In general, participants hit the ceiling buzzer 90 ms after the ball had already left the attacker's fingertips in most of the trials. This seems to reflect a strategic delay of the defensive action, most plausibly to avoid being tricked by the pump fake. The CE decreased and the number of hits (i.e., pressing the ceiling buzzer at the exact point of ball release) increased from the first to the second half of the experiment, confirming Hypothesis 3 that short-term practice with the task benefits response-precision performance. Further, the buzzer-release time (BRT) increased from the first to the second half of the experiment. As a result, the participants released the buzzer earlier, which further improved response-precision performance. The increase in response precision is not dependent on the take-off time (TOT), as the TOT was similar throughout the experiment. In contrast to the initial study [7], an additional practice block with go-trials only was added before a practice block with go-trials and stop-trials. This provided participants with a relatively longer familiarization phase prior to the six testing blocks. However, the present study had fewer trials (320) than the previous study (600) [7]. These results confirm that response precision can be improved by short-term practice in both simple and complex movements.

### Post-stop-trial adjustments

It was also predicted that responses in a go-trial would be delayed after a stop-trial. It was assumed that the larger the magnitude of the (partial) response of a failed inhibition in a stop-trial, the later participants would respond in a subsequent go-trial. This would result in less temporal precision. The results confirm Hypothesis 4, as the response-precision performance (measured by the CE) in a go-trial was reduced after a stop-trial as compared to two consecutive go-trials. The same pattern of results was found for both the BRT and the TOT. Whether participants inhibited their response successfully or not had no effect on the response-precision performance in the following go-trial. This finding aligns with the initial study [7] and other studies examining post-stop-trial adjustments in different tasks [19,20,27]. Interestingly, when

participants were only able to partially inhibit their response in a previous stop-trial, they delayed their responses the most in the present go-trial (see S1 Appendix). It may be speculated that partially inhibited responses lead to a stronger tendency to monitor the response in the following trial, further postponing the movement initiation (which results in an even larger CE in the end).

## Ground reaction forces

To provide further insight into the anticipation-response behavior and the underlying processes, ground reaction forces were measured using a force plate.

Different temporal structures were found for the force-time profiles in go-trials in which participants responded on target and for the force-time profiles in stop-trials (regardless of inhibition success). Although the movement was initiated (t0) at the same time (excepted from the go-trials compared to the successful stop-trials), t1 and t2 were reached at different times. T1 occurred earlier in go-trials than in successful stop-trials but later than in unsuccessful stop-trials. The same pattern was found for t2. Further, a (somewhat) higher force output of the response action was found when participants responded in go-trial compared to the successful and unsuccessful stop-trials. Furthermore, the maximum vertical force was reached later in go-trials than in unsuccessful stop-trials but at the same time as in successful stop-trials.

Thus, participants initiated their response movement approximately at the same time regardless of trial type (i.e., stop- or go-trials). Even if the response movement was successfully inhibited, it had been initiated nonetheless but could be aborted in the early phase of movement execution. When aligning the "corrected" PNR proper (332 ms before movement initiation), with the ground reaction force data, it can be seen that the PNR was in the early phase of the movement execution, even before the lowest velocity of the COM (t1) was reached. It follows that the inhibitory processes must be initiated in the early phase of the movement to successfully inhibit a planned action [see also 16]. Additionally, the closer the video's reversal point of the ball was to the PNR, the greater the reactive force output when response inhibition was unsuccessful (i.e., the extent of the erroneous responses increase; see S1 Appendix). This supports the assumption that inhibition processes within a complex response run successively, whereby the extent of the movement increases the later the stop-signal (i.e., video's reversal point at which the player withdraws the ball) occurs [16].

The vertical force-time profile of responses in a go-trial, separated by response-precision performance, shows that participants initiated their response movement at the same time, regardless of precision. However, from t1 onwards, there are differences in the time profile of the curve that are consistent with the precision of the responses. The highest descending velocity of the COM (t1) was reached earlier when participants responded too early, as compared to hits and too late responses. T1 was reached earlier when participants responded on target than when they responded too late. The same was found for t2, the point in time of the maximum vertical force, and the take-off time (t3). Whether the participants responded on target or too early did not influence the force output. Only too late responses seem to reduce the extent of the maximum vertical force. Thus, the temporal parameters of the force-time curves align with the response-precision performance, whereas the dynamic parameters differ. Again, it appears that participants strategically delayed their responses. An awareness of this strategy may have subsequently activated inhibitory processes that reduced the force output in order to increase the chance of inhibiting the movement when the attacker performed a pump fake.

## Practical implications and applicability

As a take home message for basketball praxis, lifting the ball up to chest height during the pump fake will result in a 50 % chance that the defender will start an inappropriate response. This likelihood can be increased by lifting the ball up to or above head height. At this point, the defender is unable to inhibit the response and will be inevitably execute the whole

body defensive blocking action. This provides the attacker with the opportunity to prepare for the (then) undefended jump shot.

## Conclusion

Motor planning processes not only seem to be concerned with the initiation of the response action, but also with the time it takes to execute the movement. In addition, inhibition processes appear to occur sequentially within complex response actions. The later the perceptual information for stopping a particular defensive action is provided, the greater the movement extent of the response action. To successfully inhibit the defensive action, the inhibition processes must already be completed in the early phase of movement execution. Moreover, responses are strategically delayed after a stop-trial, regardless of whether the inhibition was successful in the previous stop-trial.

## Supporting information

**S1 Appendix. Further analysis of post-stop-trial adjustments.**
(DOCX)

**S1 Fig. Relative frequencies (in percent) of successful and unsuccessful (partial) inhibitions.** Frequencies (in percent) of the successful inhibitions (N = 633) and of the unsuccessful inhibitions differentiated for erroneous initial responses (N = 273), and erroneous full responses (N = 343), when responding to a pump fake across the different stop-signal intervals (in ms).
(TIF)

## Author contributions

**Conceptualization:** Carolin Wickemeyer, Iris Güldenpenning, Matthias Weigelt.

**Data curation:** Carolin Wickemeyer.

**Formal analysis:** Carolin Wickemeyer, Iris Güldenpenning, Matthias Weigelt.

**Funding acquisition:** Carolin Wickemeyer, Matthias Weigelt.

**Investigation:** Carolin Wickemeyer.

**Methodology:** Carolin Wickemeyer, Iris Güldenpenning, Matthias Weigelt.

**Project administration:** Carolin Wickemeyer.

**Software:** Carolin Wickemeyer.

**Supervision:** Iris Güldenpenning, Matthias Weigelt.

**Visualization:** Carolin Wickemeyer, Iris Güldenpenning, Matthias Weigelt.

**Writing – original draft:** Carolin Wickemeyer.

**Writing – review & editing:** Carolin Wickemeyer, Iris Güldenpenning, Matthias Weigelt.

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
