## [Decision Letter · Decision Letter 0]

30 Jun 2025

Dear Dr. Wickemeyer,

Thank you for submitting your manuscript to PLOS ONE. After careful consideration, we feel that it has merit but does not fully meet PLOS ONE’s publication criteria as it currently stands. Therefore, we invite you to submit a revised version of the manuscript that addresses the points raised during the review process.

**ACADEMIC EDITOR:**

We look forward to receiving your revised manuscript.

Kind regards,

Emiliano Cè, Ph.D.

Academic Editor

PLOS ONE

Journal Requirements:

2. Thank you for stating the following financial disclosure: [I acknowledge support for the publication cost by the Open Access Publication Fund of Paderborn University with the grant number 20250103 (URL of the funder's website: https://www.ub.uni-paderborn.de/publizieren/open-access/open-access-publikationsfonds).www.ub.uni-paderborn.de/publizieren/open-access/open-access-publikationsfonds)]. 

3. In the online submission form, you indicated that your data will be submitted to a repository upon acceptance. We strongly recommend all authors deposit their data before acceptance, as the process can be lengthy and hold up publication timelines. Please note that, though access restrictions are acceptable now, your entire minimal dataset will need to be made freely accessible if your manuscript is accepted for publication. This policy applies to all data except where public deposition would breach compliance with the protocol approved by your research ethics board. If you are unable to adhere to our open data policy, please kindly revise your statement to explain your reasoning and we will seek the editor's input on an exemption.

Reviewers' comments:

Reviewer's Responses to Questions

**Comments to the Author**

1. Is the manuscript technically sound, and do the data support the conclusions?

Reviewer #1: Yes

Reviewer #2: Yes

2. Has the statistical analysis been performed appropriately and rigorously?

Reviewer #1: Yes

Reviewer #2: Yes

3. Have the authors made all data underlying the findings in their manuscript fully available?

Reviewer #1: Yes

Reviewer #2: Yes

4. Is the manuscript presented in an intelligible fashion and written in standard English?

Reviewer #1: No

Reviewer #2: No

Reviewer #1: The authors have studied the dynamics of anticipation-response inhibition in a quasi-realistic basketball scenario. The topic is interesting, particularly in the context of motor control, decision-making, and performance under pressure.

- Despite the valuable results the research offers, the manuscript needs significant language and format editing. It suffers from grammatical and syntactic inconsistencies, sometimes awkward sentence structures, and inconsistent formatting. In its current form, it does not meet the minimum level required by scientific journals.

If the authors choose to extensively revise the manuscript for clarity, grammar, and formatting, I believe it will make a valuable contribution to the field.

Reviewer #2: Dear Authors, in your manuscript The limits of action control for deceptive actions in sports: Response inhibition for the basketball pump fake you explore the spacio-temporal and dynamic conditions in inhibition performance in basketball. As mentioned, you used the quasi-realistic scenario. From the practical standpoint, and in order for your research to have an applicability in the coaching and training process, could you please explain why you halved your experiment (block 1-3 first half and block 4-6 second)? I guess it would be more applicable to do it in quarters... Moreover, your experiment lasted almost double than the real game (70-90 min vs 48 minutes), and therefore does not mimic the conditions of the basketball game realistically. Another point that needs a comment is the small sample size, consisting of both male and female sport science students. In addition, could there be any sex-related differences in the reaction time or spacio-temporal conditions that would suggest the need for separate data analysis?

The manuscript text is too long with lot of data recycled. Therefore, I would suggest to remove the redundant text (for example in the discussion part, do not repeat your aim and how it was achieved but rather go directly to discussing your results in the light what is already known. Similarly, introduction should be more general, and does not require such detailed commenting of each study in the field...

**Do you want your identity to be public for this peer review?** For information about this choice, including consent withdrawal, please see our Privacy Policy

Reviewer #1: No

Reviewer #2: No

---

## [Author Response · Author response to Decision Letter 1]

18 Aug 2025

We thank the reviewers and the editors for their thorough evaluation of our manuscript and for their constructive comments and suggestions. We have carefully considered all feedback and have revised the manuscript accordingly. Please find our detailed responses to each point below!

Journal Requirements:

Response: Thank you for bringing this to our attention. We have carefully revised the manuscript to ensure that it fully adheres to the PLOS ONE style requirements, including file naming, as outlined in the provided templates.

2. Thank you for stating the following financial disclosure: [I acknowledge support for the publication cost by the Open Access Publication Fund of Paderborn University with the grant number 20250103 (URL of the funder's website: https://www.ub.uni-paderborn.de/publizieren/open-access/open-access-publikationsfonds).www.ub.uni-paderborn.de/publizieren/open-access/open-access-publikationsfonds)]. Please state what role the funders took in the study. If the funders had no role, please state: "The funders had no role in study design, data collection and analysis, decision to publish, or preparation of the manuscript." If this statement is not correct you must amend it as needed. Please include this amended Role of Funder statement in your cover letter; we will change the online submission form on your behalf.

Response: Thank you. We have now included the following statement in our cover letter: "The funders had no role in study design, data collection and analysis, decision to publish, or preparation of the manuscript."

3. In the online submission form, you indicated that your data will be submitted to a repository upon acceptance. We strongly recommend all authors deposit their data before acceptance, as the process can be lengthy and hold up publication timelines. Please note that, though access restrictions are acceptable now, your entire minimal dataset will need to be made freely accessible if your manuscript is accepted for publication. This policy applies to all data except where public deposition would breach compliance with the protocol approved by your research ethics board. If you are unable to adhere to our open data policy, please kindly revise your statement to explain your reasoning and we will seek the editor's input on an exemption.

Response: Thank you for your comment. We have already made the data available in the OSF repository. The access link is provided both in the cover letter and in the Methods section of the manuscript as follows: "The stimulus materials and raw data can be accessed in the Open Science Framework (OSF) electronic data repository (https://osf.io/bzfp6/?view_only=0263d6f1aabe466098dfbc6c4ddc0684 )."

Response: Thank you for your comment. We have included the full ethics statement in the Methods section of the manuscript: "The Ethics Committee of Paderborn University assessed the study as ethically noncritical and granted its approval in written form (42/2023)."

Review Comments to the Author

Reviewer #1:

1. The authors have studied the dynamics of anticipation-response inhibition in a quasi-realistic basketball scenario. The topic is interesting, particularly in the context of motor control, decision-making, and performance under pressure.

- Despite the valuable results the research offers, the manuscript needs significant language and format editing. It suffers from grammatical and syntactic inconsistencies, sometimes awkward sentence structures, and inconsistent formatting. In its current form, it does not meet the minimum level required by scientific journals.

If the authors choose to extensively revise the manuscript for clarity, grammar, and formatting, I believe it will make a valuable contribution to the field.

Response: Thank you for your positive evaluation of our research and the valuable comments about the language and formatting issues. We have carefully revised the manuscript to address grammatical and syntactic inconsistencies and to improve clarity and formatting throughout. As part of this process, we utilized DeepL Write, an AI-based language editing tool, to further improve the quality of our manuscript. We hope that these changes meet the standards required by the journal and that the revision addresses your concerns appropriately.

Reviewer #2:

1. Dear Authors, in your manuscript The limits of action control for deceptive actions in sports: Response inhibition for the basketball pump fake you explore the spacio-temporal and dynamic conditions in inhibition performance in basketball. As mentioned, you used the quasi-realistic scenario. From the practical standpoint, and in order for your research to have an applicability in the coaching and training process, could you please explain why you halved your experiment (block 1-3 first half and block 4-6 second)? I guess it would be more applicable to do it in quarters... Moreover, your experiment lasted almost double than the real game (70-90 min vs 48 minutes), and therefore does not mimic the conditions of the basketball game realistically.

Response: Thank you very much for your thoughtful comment. In this experiment, our primary aim was to create a quasi-realistic response scenario by simulating defensive actions in basketball within a controlled experimental setting. We did not intend to replicate the exact time or conditional requirements of an actual basketball game. To be honest, until reading your comment, we had never thought of this practical aspect, and it sounds intriguing to do so in the future. Nevertheless, it should be noted that the mean duration of the second practice block plus the six experimental blocks was 37.8 minutes, which is closer to the actual real-game time than the total duration of the experiment. The total duration of the experiment (70-90 minutes) included the time from welcoming the participants to the farewell. Moreover, we approached the manipulation of the basketball-specific ARI task in incremental steps by first validating the stimuli and experimental procedures (cf. Wickemeyer et al., 2024), before increasing the response complexity in the present quasi-realistic scenario, to carefully investigate the underlying mechanisms of response inhibition. Nevertheless, we agree that mimicking the real-game conditions more closely, for example by aligning the experiment duration or block structure with actual basketball quarters, is an excellent suggestion and represents an important avenue for future research.

Just as a sidenote, there is a 15-minute halftime break in between the second and the third quarter in the European Basketball League, the NBA, and in the basketball tournament at the Olympics. Therefore, “dividing” the experiment for data analyses of short-term practice effects (and this is what we intended) is not that far off from a more realistic basketball game scenario.

2. Another point that needs a comment is the small sample size, consisting of both male and female sport science students. In addition, could there be any sex-related differences in the reaction time or spacio-temporal conditions that would suggest the need for separate data analysis?

Response: Thank you for your important comment. The sample size was determined based on an a priori power analysis using MorePower 6.0.4., as detailed in the “Methods” section of the manuscript. Thereby, we relied on data generated in two previous experiments to aid the power analysis to estimate the sample size.

We acknowledge that sex-differences may exist, but these were not of primary interest in the present study. In any case, the current literature is equivocal and indicates only minor sex-differences, if any. In terms of reaction time, previous studies have shown that men have slight advantages in simple reaction time and lower intra-individual variability in reaction time than women (Dykiert et al., 2012; Ghisletta et al., 2018). In addition, men tend to be more accurate in coincidence anticipation tasks (CAT). Women are more likely to underestimate the arrival time, whereas men tend to overestimate the arrival time (Sanders & Sinclair, 2010; Sanders, 2011 for a review). However, there are also null findings, which may be due to the task complexity (Sanders, 2011). Mixed or no significant differences were found regarding the influence of sex on executive functions, including inhibitory control (Gaillard et al., 2020, for a meta-analysis; Ramos-Loyo et al., 2022, for an integrative review), but only some task-specific sex differences (Gaillard et al., 2020).

However, whether there are sex-differences in the present ARI-task should be examined, as you suggested. In response to your suggestion, we conducted a series of additional analyses on the two main dependent variables for response precision (CE) and inhibition performance (PNR). Two ANOVAs were carried out with the within-subject factor 'block' and the between-subject factor 'sex' regarding response precision (CE) and response inhibition (PNR). We found no significant main effect for the factor ‘sex’ and no significant interaction between the factors. Thus, there were no sex-differences for the main dependent variables in the primary analyses.

Another ANOVA was conducted for secondary sequential effects, with the within-subject factor 'trial n-1' and the between-subject factor 'sex' regarding response-precision (CE). We found a significant interaction between the factors. The results show that men and women do not differ significantly in terms of their response precision. However, females’ response precision was worse in a go-trial after a successful stop-trial [t(9) = -7.064, p < .001, d = -2.234] and after an unsuccessful stop-trial [t(9) = -9.165, p < .001, d = -2.898], compared to the response-precision after a go-trial. The response-precision in a go-trial after a successful stop-trial and after an unsuccessful stop-trial was not different. The results are in line with the overall findings in the present paper regarding post-stop-trial adjustments. Males’ response precision in a go-trial after a successful stop-trial was similar compared to the response precision after a go-trial (p = .377). The response precision in a go-trial after an unsuccessful stop-trial was worse compared to the response precision after a go-trial [t(10) = -4.503, p = .001, d = -1.358] and compared to the response precision after a successful stop-trial [t(10) = -3.071, p = .012, d = -.926]. The response-precision in a go-trial after a consecutive go-trial and after a successful stop-trial did not differ.

In sum, there were no sex-differences for the two main dependent variables regarding response precision and inhibition performance in the primary analyses. There was a single difference in the secondary analysis regarding the sequential effect of males in a go-trial after an unsuccessful stop-trial, whereas any other comparison was similar between females and males. Therefore, and due to the limited sample size in our study, separate analyses by sex do not seem to yield reliable results, which bare information on potential sex-differences. Neither can the present results be taken to suggest the absence of such sex-differences for similar reasons. However, we agree that investigating sex-related differences in domain-specific ARI-tasks is an interesting avenue for future studies with larger samples.

3. The manuscript text is too long with lot of data recycled. Therefore, I would suggest to remove the redundant text (for example in the discussion part, do not repeat your aim and how it was achieved but rather go directly to discussing your results in the light what is already known. Similarly, introduction should be more general, and does not require such detailed commenting of each study in the field...

Response: Thank you for your helpful feedback. We have shortened the manuscript and removed redundant text, particularly in the Introduction and Discussion sections, as suggested. We hope that these revisions address your concerns and improve the clarity and conciseness of the manuscript.

This concludes our point-by-point response to the reviewer’s comments. Again, we thank you for your valuable feedback and look forward to the further handling of our manuscript.

---

## [Decision Letter · Decision Letter 1]

5 Sep 2025

The limits of action control for deceptive actions in sports: Response inhibition for the basketball pump fake

PONE-D-25-24813R1

Dear Dr. Wickemeyer,

We’re pleased to inform you that your manuscript has been judged scientifically suitable for publication and will be formally accepted for publication once it meets all outstanding technical requirements.

Kind regards,

Emiliano Cè, Ph.D.

Academic Editor

PLOS ONE

Additional Editor Comments (optional):

Reviewer #1:

Reviewers' comments:

Reviewer's Responses to Questions

**Comments to the Author**

Reviewer #1: All comments have been addressed

2. Is the manuscript technically sound, and do the data support the conclusions?

Reviewer #1: Yes

3. Has the statistical analysis been performed appropriately and rigorously?

Reviewer #1: Yes

4. Have the authors made all data underlying the findings in their manuscript fully available?

Reviewer #1: (No Response)

5. Is the manuscript presented in an intelligible fashion and written in standard English?

Reviewer #1: Yes

Reviewer #1: Thank you for answering my comments and revising the manuscript accordingly. Now the manuscript looks fine to me and I can suggest its publication.

**Do you want your identity to be public for this peer review?** For information about this choice, including consent withdrawal, please see our Privacy Policy

Reviewer #1: No

---

## [Editor Report · Acceptance letter]

PONE-D-25-24813R1

PLOS ONE

Dear Dr. Wickemeyer,

I'm pleased to inform you that your manuscript has been deemed suitable for publication in PLOS ONE. Congratulations! Your manuscript is now being handed over to our production team.

Kind regards,

on behalf of

Prof. Emiliano Cè

Academic Editor

PLOS ONE